# Silencing LncRNA *CASC9* inhibits proliferation and invasion of colorectal cancer cells by MiR-542-3p/ILK

Heping Zhang[1]☉, Jingfang Wang[2]☉, Taoyuan Yu[3], Jingmin Wang[4], Jun Lu[5], Zongyang Yu[6]*

**1** Department of Anorectal, People's Hospital of Jiaozuo, Jiaozuo, Henan Province, China, **2** Medical College of Rehabilitation, Fujian University of Traditional Chinese Medicine, Fuzhou, Fujian Province, China, **3** Institute of International Education, Beijing University of Chemical Technology, Beijing, China, **4** Infertility Clinic, People's Hospital of Jiaozuo, Jiaozuo, Henan Province, China, **5** Basic Medical Laboratory, 900th Hospital of the Joint Logistics Team, Fuzhou, Fujian Province, China, **6** Pulmonary and Critical Care Medicine, 900th Hospital of the Joint Logistics Team, Fuzhou, Fujian Province, China

☉ These authors contributed equally to this work.
* yuzy527@sina.com

**Data Availability Statement:** All relevant data are within the paper and its Supporting Information files.

**Funding:** The authors received no specific funding for this work.

## Abstract

Colorectal cancer (CRC) ranks the third in cancers and the second in the reasons of cancer-related death. More evidence indicates that long non-coding RNA participates in tumor initiation and progression. It's known that cancer susceptibility candidate 9 is an oncogenic long non-coding RNA in CRC. miR-542-3p is a negative regulator of CRC, while integrin-linked kinase could contribute to tumor progression and chemoresistance. However, the correlation among long non-coding RNA cancer susceptibility candidate 9, miR-542-3p and integrin-linked kinase in CRC is still unclear. We demonstrated long non-coding RNA cancer susceptibility candidate 9 in CRC specimens and cell lines overexpressed via real-time quantitative polymerase chain reaction. Once long non-coding RNA cancer susceptibility candidate 9 was knocked down, it significantly inhibited proliferation, invasion, and migration of CRC cells in real-time quantitative polymerase chain reaction, cell counting kit-8, 5-ethy-nyl-2'-deoxyuridine, and transwell assays, which also was validated *in vivo*. Long non-coding RNA cancer susceptibility candidate 9 negatively regulates miR-542-3p in a targeted manner, and the function of up-regulated miR-542-3p was confirmed similarly. While miR-542-3p negatively regulates integrin-linked kinase. Thus, we further verified that overexpression of integrin-linked kinase on down-regulated long non-coding RNA cancer susceptibility candidate 9 or up-regulated miR-542-3p significantly restored CRC cell proliferation via bioinformatic analysis, dual-luciferase report assay, real-time quantitative polymerase chain reaction, RNA immunoprecipitation, and western blot. This study testified that silencing long non-coding RNA cancer susceptibility candidate 9 could inhibit proliferation and invasion of CRC cells by miR-542-3p/integrin-linked kinase.

**Competing interests:** The authors have declared that no competing interests exist.

## Introduction

Colorectal cancer (CRC) ranks the third in cancer and the second in the reason of cancer-related death worldwide. Increasing evidence indicates that long non-coding RNA (lncRNA) participates in tumor initiation and progression [1, 2]. LncRNA is a special non-coding RNA molecule of more than 200 nucleotides in length and has been known as a potential novel prognostic biomarker in CRC [2]. It's known that as an oncogenic lncRNA [3]. Cancer suscep-tibility candidate 9 (*CASC9*) is usually high in CRC and associated with advanced tumor-node-metastasis (TNM) stage and poor prognosis. A study identified the *CASC9*/miR-576-5p/AKT3 axis is relevant to the proliferation and apoptosis of CRC [4]. Knockdown of *CASC9* inhibits the proliferation and migration through AKT/mTOR/EMT signaling [5] and pro-motes apoptosis in CRC cells [3]. In summary, lncRNA *CASC9* which plays the role of an oncogene is abnormally highly expressed in CRC.

MiR-542-3p has been reported to play a role in tumor suppressor genes and is low expressed in CRC [6–8].

According to the report, integrin-linked kinase (ILK) is a target gene of miR-542-3p in oste-osarcoma, and upregulating miR-542-3p will downregulate ILK [9]. In CRC, overexpression of ILK is associated with genomic instability, epithelial-mesenchymal transition (EMT), and cancer stem cells (CSCs) traits, which contribute to tumor progression, migration, invasion, and chemoresistance via *NF-κB*, *CTEN* signals, etc. [10–15]. High immunohistochemical expression of ILK is associated with poor prognostic and pathological parameters [15, 16].

Based on the above, we hypothesized that up-regulating lncRNA *CASC9* down-regulate miR-542-3p, overexpress ILK and participate in CRC as a result. In this review, silencing lncRNA *CASC9* could inhibit CRC by regulating miR-542-3p/ILK.

## Materials and methods

### Study design

The study was designed to characterize the inhibiting effect of silencing lncRNA *CASC9* on CRC by miR-542-3p/ILK. Based on the literature, we first hypothesized that up-regulating lncRNA *CASC9* could down-regulate miR-542-3p, overexpress ILK, at last participate in CRC. In this study, human normal intestinal epithelial cells and two human CRC cells were used *in vitro* assays. 36 samples were independently derived from at least 36 unrelated patients. We first detected the expression of lncRNA *CASC9*, then evaluated the effect of lncRNA *CASC9* on CRC, and explored how miR-542-3p/ILK affected lncRNA *CASC9*. To translate the findings *in vitro* into *in vivo*, we designed an animal experiment. All studies were performed in technical duplicates or triplicates as indicated. Results shown were either representative of or mean of at least three independent experiments (otherwise annotated in the legends).

**CRC samples analysis.** With patients' written informed consent, thirty-six paired fresh primary colorectal cancer and matched adjacent normal tissues were collected by surgical from adult CRC patients and were immediately frozen in a −80˚C freezer. Samples were obtained from September 2020 to March 2021 and detected in April 2021. Authors had no access to information that could identify individual participants during or after data collection. This study was approved by the Biomedical Ethics Committee of the 900th Hospital of the Joint Logistics Team (Ethics approval number 2020–023).

**Cell culture.** Human normal intestinal epithelial cells NCM460, and CRC cell lines HCT116 and SW620 were obtained from the American Type Culture Collection (Manassas, VA, USA). HCT116, SW620, and NCM460 cells were maintained in DMEM or RPMI (Gibco, Grand Island, NY, USA). The mediums were supplemented with 10% fetal bovine serum

(FBS) (Gibco) and 1% penicillin-streptomycin mixture. The cells were cultured at 37°C in a 5% $CO_2$ incubator. All cell lines were free of mycoplasma and were authenticated by genetic profiling using polymorphic short tandem repeat loci.

**Stable *CASC9*-knockdown cell lines.** Lentiviral particles were produced in HEK293T cells and co-transfected with sh-*CASC9*–1, sh-*CASC9*–2, sh-*CASC9*–3, or corresponding empty vectors. Next, HCT116 and SW620 were infected with lentiviral particles in the presence of 8 μg/ml polybrene (Genechem, Shanghai, China) and selected using 0.5 μg/ml puromycin. Puromycin-resistant cell pools were collected and verified by qPCR one week later.

**Real-time quantitative polymerase chain reaction (qPCR).** Total RNA was extracted using TRIzol reagent and was treated with RQ1 RNase-Free DNase (Promega, Madison, WI, USA) for 30 minutes. cDNA was synthesized using the M-MLV reverse transcription kit (Promega) as manufacturer's instructions. ChamQ Universal SYBR qPCR Master Mix (Vazyme Biotech, Nanjing, China) on a Quant Studio 3 instrument (Thermo Fisher Scientific) was used to conduct qPCR. GAPDH and U6 were used as the internal reference of lncRNA *CASC9* and miR-542-3p, respectively. The expression of targeted genes was calculated using $2^{-\Delta\Delta Ct}$.

**CCK-8 assay for cell proliferation.** Firstly, $2 \times 10^3$ cells/well were seeded into a 96-well plate. After attachment, cells were transfected with siRNA and incubated in an incubator at 37°C with 5% $CO_2$. After 24, 48, 72, and 96 hours, the medium containing 10% CCK-8 (Meilun Biotechnology, Dalian, China) was placed into per well. The cells were incubated at 37°C for 30 minutes [4]. Then, the absorbance was detected at a wavelength of 450 nm with a microplate reader (Tecan, Lyon, France).

**EdU assay for cell proliferation.** A BeyoClick™ EdU-488 Cell Proliferation Kit (Beyotime, C0071S) was used to measure cell proliferation as manufacturer's instructions. $4 \times 104$ HCT116 and SW620 cells/well were seeded into plates and cultured for 24 hours, then were nurtured with 10 μM EdU for 2 hours, followed by fixing with 4% paraformaldehyde (Beyotime, P0099) for 15 minutes and treatment of 0.3% Triton X-100 (Beyotime) for 10 minutes. Afterward, cells were incubated with the click reaction solution for 30 minutes. The nuclei were stained with Hoechst 33342 for 10 minutes [17]. Cells were observed with a fluorescence microscope (Olympus, Tokyo, Japan). The ratio of Edu-positive cells (Edu-positive cells/ Hoechst 33342-stained cells) was calculated to analyze the proliferative ability.

**Transwell assay for cell invasion and migration.** 24-well transwell coated with Matrigel (8-μm pore size; BD Biosciences, San Jose, CA) were used for invasion and migration experiments. $1 \times 10^5$ non-transfected cells or stably transfected cells were plated on separate wells and were cultured overnight in serum-free medium before trypsinization and re-suspended at a density of $2 \times 10^5$ cells/ml in DMEM including 1% FBS. The cells were loaded to the upper chamber, with DMEM including 10% FBS as a chemoattractant in the lower chamber. The DMEM including 1% FBS in the lower chamber was used as a control. The Matrigel and the cells remaining in the upper chamber were 24 hours incubation. The cells in the lower surface of the membrane were stained with hematoxylin after being fixed with formaldehyde solution. The cells in at least five random microscopic fields (×200) were counted and photographed [8].

**Luciferase reporter gene assay for genes' targeted interaction.** To generate luciferase reporter constructs, the dual-luciferase miRNA target expression vector pmirGLO (Promega, Madison, Wisconsin, USA) was used. Full length of lncRNA *CASC9* or ILK 3′-UTR sequence with wild-type (WT) and mutant type (Mut) miRNA binding sites were obtained from Vigene Biosciences (Rockville, Maryland). Cells were seeded into 96-well plates and co-transfected with wild-type or mutant lncRNA *CASC9* or ILK 3′-UTR constructs and miR-542-3p mimic using Lipofectamine 3000. After 48 hours, luciferase activity was measured using the dual-luciferase reporter assay system (Promega) and assessed.

*In vivo* **assay and xenograft tumor analysis.** 6-week-old male BALB/c nude mice were purchased from Shanghai Laboratory Animal Center (Shanghai, China) and housed under pathogen-free conditions. Logarithmically growing cells were harvested and resuspended in PBS. $1 \times 10^7$ SW620 or *CASC9*-knockdown SW620 cells were subcutaneously injected into the rear flanks of a mouse. Xenograft tumors were measured by a Vernier caliper, and tumor volume was calculated using the following equation: $V = L \times W^2 \times 0.5236$ (L = long axis, W = short axis) every week until the fourth week.

**Immunohistochemistry.** Paraffin sections (5 μm) from xenograft tumor samples were deparaffinized in 100% xylene and rehydrated with a decreasing ethanol series and then water. Antigen retrieval was conducted in 0.01 mol/L sodium citrate buffer (pH 6.0) at 100˚C for 10 minutes. IHC assay was performed as described previously using supersensitive horseradish peroxidase immunohistochemistry kit (Sangon Biotech, Shanghai, China) [3] per manufacturer's instructions with anti-Ki67 antibody (sc-23900, Santa Cruz) and anti-MMP2 antibody (sc-13595, Santa Cruz).

**RNA immunoprecipitation assay.** A Magna RNA immunoprecipitation (RIP) Kit (Geneseed, Guangzhou, China) was used as the manufacturer's instructions. Antibodies for RIP assay of Ago2 and control IgG were from Abcam. The co-precipitated RNAs were detected by qPCR. The total RNAs were the input controls.

**Western blot analysis.** Cells were lysed with RIPA buffer, and total protein was quantified using the BCA Protein Assay Kit (Beyotime). Total denatured protein (30 μg) was subjected to sodium dodecyl sulfate-polyacrylamide gels and transferred to polyvinylidene fluoride membranes (Millipore). The blots were incubated with primary antibodies ILK (ab76468, 1:5000 dilution, Abcam, USA) and β-actin (ab8227,1:5000 dilution, Abcam, USA) at 4˚C overnight. The next morning, after incubated at room temperature for 1 hour, the blots were washed 30 minutes with TBST, then incubated with horseradish peroxidase-linked secondary antibody at room temperature for 1 hour. After being washed with TBST and PBS, immunocomplexes were detected with Super Signal West Pico Chemiluminescent Substrate (Thermo Fisher Scientific).

## Statistical analysis

Data were presented as the mean ± standard deviation (SD). and were analyzed in GraphPad Prism 8.0 (GraphPad Software, La Jolla, CA, USA). Statistical significance was analyzed by two-tailed Student's t-test, Mann Whitney U-test, or Wilcoxon signed-rank test. Differences between groups were determined using two-way ANOVA. The significance of the clinicopathologic parameters of CRC patients was determined by the chi-square test. All P values were two-tailed, and $P < 0.05$ was considered statistically significant.

## Results

### LncRNA *CASC9* expresses highly in CRC

The expression of lncRNA *CASC9* in CRC specimens and two cell lines of CRC were firstly evaluated. The 36 specimens (carcinomas and para-cancerous tissues) were obtained from CRC patients and detected via qPCR. LncRNA *CASC9* of carcinomas significantly expressed more highly than that of para-cancerous tissues in CRC patients (p<0.01) (Fig 1A). We further analyzed whether there was a difference between TNM I&II and TNM III. We found that the expression of lncRNA *CASC9* in TNM III was more significant than those in TNM I&II, which declared that the expression of lncRNA *CASC9* was positively related to the TNM stage (Fig 1B). In *vitro*, compared to NCM460, lncRNA *CASC9* overexpressed both in SW620 and

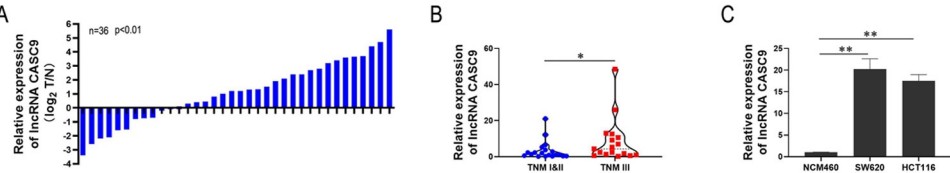

**Fig 1. LncRNA *CASC9* expresses highly in CRC.** (A)The relative expression of lncRNA *CASC9* of carcinomas and para-cancerous tissues from patients with CRC was determined by qPCR. Histogram of relative expression of lncRNA *CASC9* ($\log_2$T/N) from 36 patients(n = 36), one histogram represents one specimen. Most of these values are positive (p<0.01). (B) Subsets were defined as TNM I&II and TNM III from 36 carcinomas, relative expression of lncRNA *CASC9* in two subsets were compared, and the expression in TNM III was significantly higher than that in TNM I&II. Each dot represents one carcinoma (n = 36). (C) Relative expression of lncRNA *CASC9* in NCM460, SW620, and HCT116 are determined. Compared to NCM460, lncRNA *CASC9* overexpressed both in SW620 and HCT116 cell lines. $^*$P < 0.05 and $^{**}$P < 0.01.

HCT116 cell lines, which was coincident with the result of human tissues (Fig 1C). Therefore, we conclude that lncRNA *CASC9* expresses highly in CRC.

## Downregulating lncRNA *CASC9* inhibits the proliferation and invasion of CRC cells

To characterize the function of lncRNA *CASC9*, we turned to the *vitro* system. We firstly constructed three lentiviral vectors with lncRNA *CASC9* knocked down, packaged lentivirus and infected CRC cells. Three stable transformants of CRC cell lines were obtained with down-regulation of lncRNA *CASC9*, which were identified by qPCR. The expression of lncRNA *CASC9* on both SW620 and HCT116 cell lines confirmed the same pattern as observed. Compared to the control subset, 3 experimental subsets all could reduce the expression of lncRNA *CASC9*, especially in sh-*CASC9*-2 subset (Fig 2A). Thus, in the following assay, we used sh-*CASC9*-2 to downregulate lncRNA *CASC9*. The proliferation of both SW620 and HCT116 with sh-NC and sh-*CASC9*-2 were tested by CCK8 and showed that knocking down lncRNA *CASC9* could significantly inhibit the proliferation of CRC (Fig 2B and 2C). Next, EdU was used as double labeling to detect the newly synthesized DNA of SW620 and HCT116 cells for detecting the cell proliferation rate. The results indicated that knocking down lncRNA *CASC9* significantly inhibited proliferation (Fig 2D). Transwell experiments showed that knocking down lncRNA *CASC9* significantly inhibited the invasion and migration of SW620 and HCT116 (Fig 2E and 2F). In summary, downregulation of lncRNA *CASC9* inhibits the proliferation and invasion of CRC.

## Downregulating lncRNA *CASC9* in SW620 xenograft tumor inhibits SW620 proliferation

To explore the functional consequences of downregulating lncRNA *CASC9* in SW620 xenograft tumor, SW620 cells with stable down-regulation of lncRNA *CASC9* were inoculated into nude mice to establish xenograft tumor models. Growth curves of xenograft tumors were drawn and weights of xenograft tumors were measured. After analyzing, we knew that downregulation of lncRNA *CASC9* could significantly inhibit xenograft tumor growth (Fig 3A). Weights of transplanted tumors in the experimental group were much lighter than those of the control group (Fig 3B). Ki67 and MMP2 were detected through IHC, and Ki67 reduced significantly in positive cells (Fig 3C), while the decrease of MMP2 in positive cells was not so obvious as Ki67 (Fig 3D). The expression levels of lncRNA *CASC9*, miR-542-3p, and ILK on xenograft CRC were tested via qPCR. The results showed that lncRNA *CASC9* and ILK were

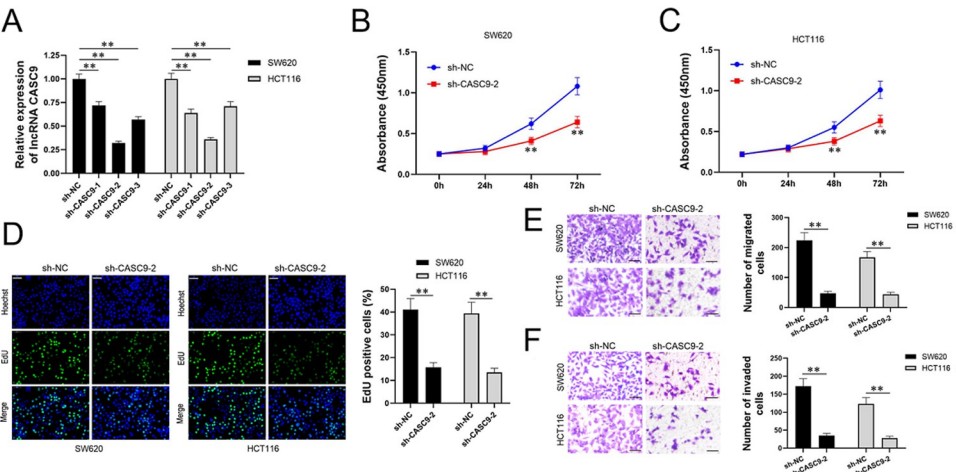

**Fig 2. Downregulating lncRNA *CASC9* inhibits proliferation and invasion of CRC cells.** Relative expression of lncRNA *CASC9* in SW620 and HCT116 cell lines of CRC were determined by qPCR, Black represents SW620 and grey represents HCT116 in histograms. (A)Subsets were defined as sh-NC, sh-*CASC9*-1, sh-*CASC9*-2, and sh-*CASC9*-3. sh-*CASC9*-2 downregulated the expression of lncRNA *CASC9* most significantly. (B&C) Then, we respectively detected the proliferation of SW620 and HCT116 on sh-NC and sh-*CASC9*-2 subsets at 0, 24, 48, and 72h via CCK8, the proliferation of both SW620 and HCT116 with sh-*CASC9*-2 were significantly slower than the sh-NC subset, and the inhibitory effect was in a time-dependent manner. Blue represents the sh-NC subset, and red represents the sh-*CASC9*-2 subset. (D) EdU assay indicated that knocking down lncRNA *CASC9* significantly inhibited the proliferation of CRC. Histograms of Edu positive cells on sh-NC and sh-*CASC9*-2 subsets are quantized results of left pictures. (E&F) Transwell was tested for invasion and migration of SW620 and HCT116 cell lines on sh-NC and sh-*CASC9*-2 subsets. Sh-*CASC9*-2 subsets showed significant inhibition of invasion and migration. The upper picture is about migration, and the below one is about invasion. The histograms are the quantitative analysis of the left pictures. Scale bar = 50 μm. *P < 0.05 and **P < 0.01.

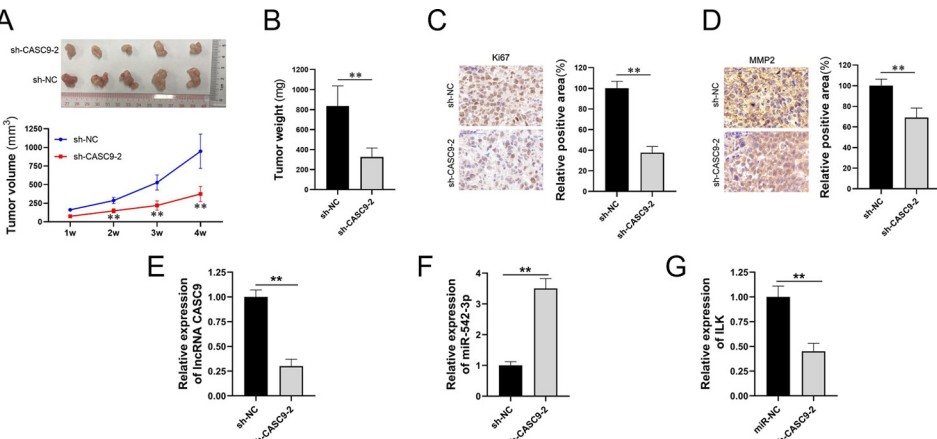

**Fig 3. Downregulating lncRNA *CASC9* in SW620 xenograft tumor inhibits SW620 proliferation.** SW620 with stable down-regulation of lncRNA *CASC9* were inoculated into nude mice to establish xenograft tumor models. Two groups were designed, sh-NC was the normal group, (i.e., control group), and the other one was an experimental group with which stable down-regulation of lncRNA *CASC9* xenograft tumor. Black represents sh-NC and grey represents sh-*CASC9*-2 in histograms. (A)Growth curves of xenograft tumors were drawn respectively in 1, 2, 3, 4 weeks, and tumor volume was inhibited in the sh-*CASC9*-2 subset. The blue represents sh-NC and red represents sh-*CASC9*-2 in the line chart. (B)Tumor weights were detected at last, and the tumor in the sh-*CASC9*-2 subset was lighter than sh-NC. Ki67 and MMP2 were detected by IHC. (C) Ki67 in positive cells significantly reduced in the sh-*CASC9*-2 subset, (D) while the decrease of MMP2 in positive cells was not so obvious. The histograms are the quantitative analysis of pictures. (E&F&G) Compared to the NC subset, the expression of lncRNA *CASC9* and ILK in the sh-CASC9-2 subset was significantly downregulated, while the expression of miR-542-3p in the sh-CASC9-2 subset was significantly upregulated. Scale bar = 50 μm. *P < 0.05 and **P < 0.01.

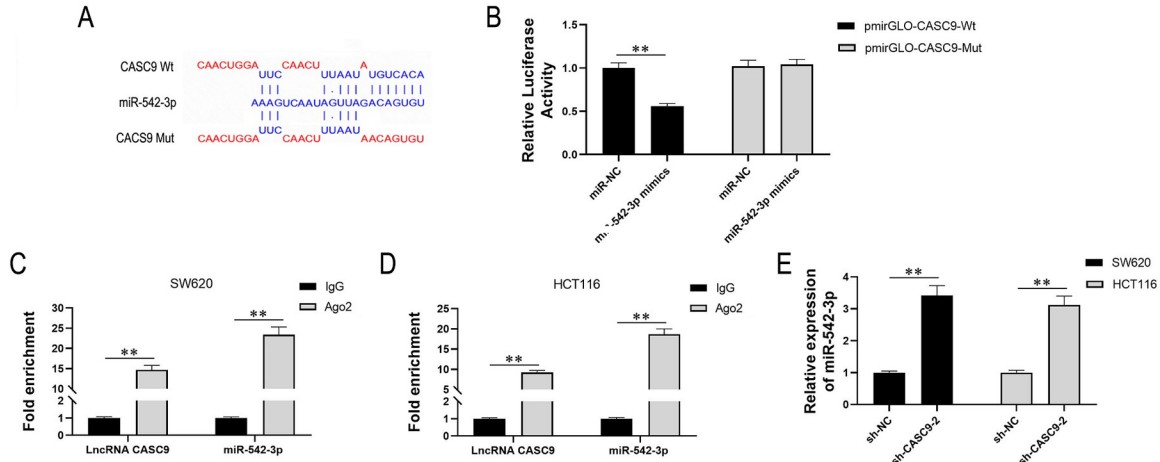

**Fig 4. LncRNA *CASC9* negatively regulates miR-542-3p by molecular sponge adsorption.** (A) the binding sites for miR-542-3p in lncRNA *CASC9* were prompted by bioinformatics analysis. (B) Dual-luciferase reporter gene assay showed that that the luciferase activity of the Wt reporter in the lncRNA CASC9-Wt+miR-542-3p mimics subset was significantly decreased compared to 3 other subsets. Black represents pmirGLO-*CASC9*-Wt and grey represents pmirGLO-*CASC9*-Mut in histograms. (C&D) In RNA-IP assay, the fold enrichment in the Ago2 subset was significantly higher than the IgG subset, which further confirmed lncRNA *CASC9* could combine miR-542-3p by predicted binding sites both in SW620 and HCT116. Mouse IgG and human Ago2 were used in lncRNA *CASC9* and miR-542-3p subsets of CRC cell lines. (E)The relative expression of miR-542-3p of both NC and sh-*CASC9*-2 in SW620 and HCT116 were detected via qPCR, and downregulating lncRNA *CASC9* upregulated miR-542-3p. Black represents SW620 and grey represents HCT116 in histograms. $^*$P < 0.05 and $^{**}$P < 0.01.

downregulated and miR-542-3p was upregulated (Fig 3E–3G), which suggests that there may be a correlation among lncRNA CASC9, miR-542-3p and ILK. The *in vivo* experiment indicated that downregulating lncRNA *CASC9* could inhibit CRC from proliferating and may reduce its invasion.

## LncRNA *CASC9* negatively regulates miR-542-3p by molecular sponge adsorption

Firstly, we predicted the binding sites for miR-542-3p in lncRNA CASC9 via bioinformatics analysis (Fig 4A). Then, dual-luciferase reporter gene assay with lncRNA *CASC9* plasmid vector including Wt and Mut were designed. The results showed that the luciferase activity of the Wt vector in the lncRNA *CASC9*-Wt+miR-542-3p mimics subset was significantly reduced compared with the lncRNA *CASC9*-Wt+NC subset but had no effect on that of Mut vector (Fig 4B). In the RIP assay, the fold enrichment in the Ago2 subset was significantly higher than the IgG subset, which further confirmed lncRNA *CASC9* could combine miR-542-3p both in SW620 and HCT116 (Fig 4C and 4D). Besides, miR-542-3p in SW620 and HCT116 with down-regulated lncRNA *CASC9* were observed the same pattern. MiR-542-3p was significantly up-regulated in the downregulation of lncRNA *CASC9* cells. Thus, lncRNA *CASC9* negatively regulated miR-542-3p (Fig 4E). At last, we summarize that lncRNA *CASC9* negatively regulates miR-542-3p in a targeted manner in CRC.

## Up-regulating miR-542-3p inhibits the proliferation and invasion of CRC cells

To study the effect of up-regulated miR-542-3p on CRC, we transfected miR-542-3p mimic into SW620 and HCT116 cells and identified the expression of miR-542-3p by qPCR. MiR-

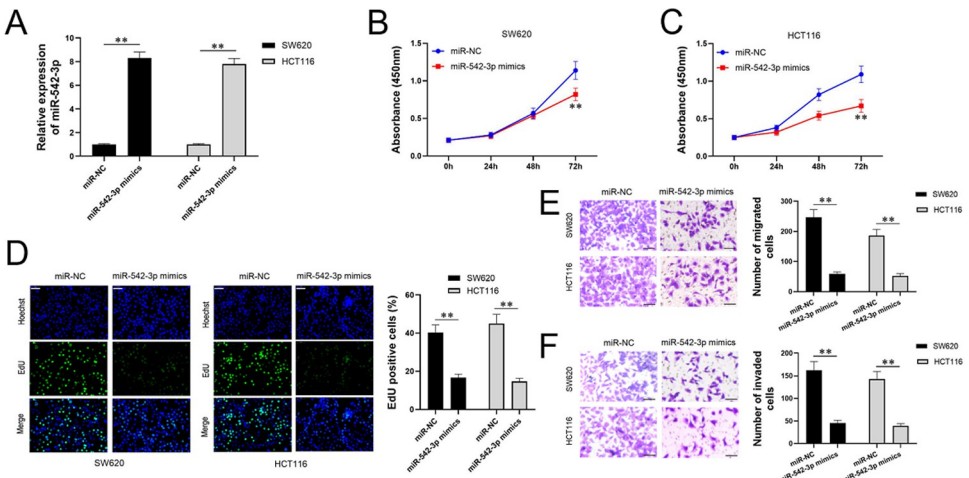

**Fig 5. Up-regulating miR-542-3p inhibits the proliferation and invasion of CRC cells.** Relative expression of miR-542-3p mimics in SW620 and HCT116 cell lines of CRC were determined by qPCR, Black represents SW620 and grey represents HCT116 in histograms. (A) Subsets were defined as miR-NC and miR-542-3p mimics. MiR-542-3p mimics did upregulate the expression of miR-542-3p significantly. (B&C) On the initial results, we respectively detected the proliferation of SW620 and HCT116 on miR-NC and miR-542-3p mimics subsets at 0, 24, 48, and 72h via CCK8, and found that SW620 with miR-542-3p mimics proliferated significantly slower than miR-NC subset after 72 hours, while HCT116 with miR-542-3p mimics proliferated significantly slower than miR-NC subset after 48 hours. Blue represents the miR-NC subset, and red represents miR-542-3p mimics. (D) Edu assay showed miR-542-3p mimics significantly inhibited CRC from proliferating. Histograms of Edu positive cells in miR-NC and miR-542-3p mimics subsets are the quantitative analysis of pictures. (E&F) Transwell was used for the invasion and migration of SW620 and HCT116 cell lines. Both invasion and migration were inhibited in miR-542-3p mimics subsets. The upper picture is about migration, and the below one is about invasion. The histograms are the quantitative analysis of the left pictures. Scale bar = 50 μm. *P < 0.05 and **P < 0.01.

542-3p mimic did upregulate miR-542-3p significantly (Fig 5A). Next, the effect of up-regulated miR-542-3p on proliferation was detected using CCK8. The results showed SW620 with up-regulated miR-542-3p proliferated significantly slower than the miR-NC subset after 72 hours, which appeared earlier in HCT116. (Fig 5B and 5C). In the EdU assay, miR-542-3p mimics significantly inhibited CRC cells, which meant that up-regulating miR-542-3p significantly inhibits CRC from proliferating (Fig 5D). Transwell experiments showed that both invasion and migration of SW620 and HCT116 cells were inhibited by miR-542-3p mimics, which indicated up-regulating miR-542-3p significantly inhibited the invasion and migration of CRC (Fig 5E and 5F). In addition, the effect of upregulating miR-542-3p is consistent with downregulating lncRNA *CASC9* in CRC cells.

## ILK is the target gene of miR-542-3p

We analyzed the bioinformatics and predicted the binding sites for miR-542-3p in ILK 3 'UTR (Fig 6A). Dual-luciferase reporter gene assay with ILK plasmid vector including Wt and Mut 3' UTR were designed. The results showed that the luciferase activity of the Wt reporter in the ILK-Wt+miR-542-3p mimics subset was significantly reduced compared with the ILK-Wt +NC subset while not affecting that of the Mut vector (Fig 6B). In protein level, ILK was down-regulated in miR-542-3p mimics in SW620 and HCT116. Besides, compared to the control groups, the difference was significant. Thus, miR-542-3p negatively regulated ILK (Fig 6C). Accordingly, ILK is the target gene of miR-542-3p, and miR-542-3p negatively regulates ILK.

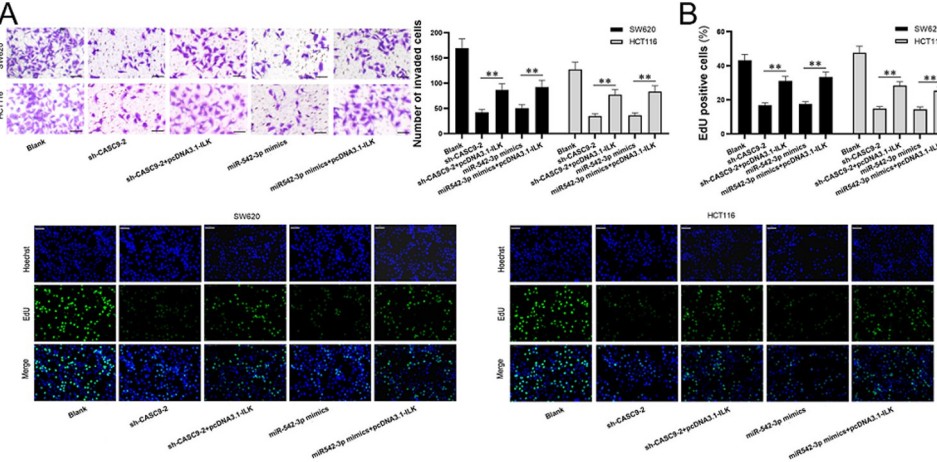

**Fig 6. ILK is the target gene of miR-542-3p.** (A) We first found the binding sites for miR-542-3p in ILK 3 'UTR via bioinformatics analysis, (B) then ILK 3 'UTR -Wt and ILK 3 'UTR -Mut were designed, and the results of dual luciferase report assay revealed that the luciferase activity of the Wt reporter in the ILK-Wt+miR-542-3p mimics subset was significantly reduced compared to 3 other subsets. (C) The effect of transfected miR-542-3p mimic on the expression of the ILK protein was detected by WB. ILK protein was downregulated in miR-542-3p mimics subsets both in SW620 and HCT116. Three subsets were designed and compared to blank or miR-NC, ILK in up-regulated miR-542-3p expressed lower. Black represents pmirGLO-ILK 3 'UTR -Wt and grey represent pmirGLO-ILK 3 'UTR -Mut in histograms. The right picture is the quantitative analysis of the left one. Black represents SW620 and grey represents HCT116 in histograms. $^*P < 0.05$ and $^{**}P < 0.01$.

## Overexpression of ILK significantly restores the proliferation and invasion of CRC cells inhibited by downregulated lncRNA *CASC9* or upregulated miR-542-3p

Finally, we wanted to know miR-542-3p/ILK axis is indispensable for downregulating lncRNA *CASC9* in CRC cells. The response of down-regulation of lncRNA *CASC9* or up-regulation of miR-542-3p to overexpression of ILK were detected via transwells and Edu assays. We confirmed that both down-regulating lncRNA *CASC9* or up-regulating miR-542-3p could inhibit the invasion, while overexpression of ILK could significantly restore part of the invasion inhibited by downregulated lncRNA *CASC9* or upregulated miR-542-3p, which existed not only in SW620 but also in HCT116 (Fig 7A). EdU assay verified that overexpression of ILK

**Fig 7. Overexpression of ILK significantly restores the proliferation and invasion of CRC cells with downregulated lncRNA *CASC9* or upregulated miR-542-3p.** 5 groups were designed, they are blank, sh-*CASC9*-2, sh-*CASC9*-2+pcDNA2.1-ILK, miR-542-3p mimics, and miR-542-3p mimics+pcDNA3. Black represents SW620 and grey represents HCT116 in histograms. (A) Transwell was used for invasion on 5 subsets of SW620 and HCT116 cell lines. Both down-regulating lncRNA *CASC9* and up-regulating miR-542-3p could inhibit the invasion, while overexpression of ILK could significantly restore part of proliferation which is inhibited by downregulated lncRNA *CASC9* and upregulated miR-542-3p, not only in SW620 but also in HCT116. (B) Edu assay was determined by comparing the proliferation among 5 groups. Compared to down-regulating lncRNA *CASC9* and up-regulating miR-542-3p, overexpression of ILK significantly improved the proliferation inhibited by downregulated lncRNA *CASC9* and upregulated miR-542-3p. Histograms are the quantitative analysis of pictures. Scale bar = 50 μm. $^*P < 0.05$ and $^{**}P < 0.01$.

significantly improved the proliferation which was inhibited by down-regulated lncRNA *CASC9* or up-regulated miR-542-3p (Fig 7B). Thus, overexpression of ILK can significantly weaken the inhibition on proliferation and invasion which is inhibited by downregulated lncRNA *CASC9* and upregulated miR-542-3p in CRC.

## Discussion

*CASC9* is mainly concentrated in the cytoplasm and significantly upregulated in human tumor tissues including CRC. ILK, located in the nucleus and centrosomes, is abnormally overexpressed in CRC tissues. We compared the survivals of *CASC9* and ILK using the GEPIA database (http://gepia.cancer-pku.cn/) and found that both *CASC9* and ILK had the similar tendency that the survival time of the low gene expression group was longer than that of the high gene expression group over time, while the tendency in miR-542-3p was not obvious. It suggests that there may be some correlation between *CASC9* and ILK. Some studies found that as a competing endogenous RNA, *CASC9* inhibits miRNA expression by competitively binding to its target miRNA at the 3′UTR region via a sponging effect and then regulates the progression of various human neoplasms [4, 18]. The expression of miR-542-3p in CRC tissues was verified abnormally downregulated [6, 19, 20], which was verified in our study as well. Some researchers have demonstrated miR-542-3p overexpression down-regulating ILK then inhibiting osteosarcoma has been verified [9]. Thus, there may be a potential mechanism that *CASC9* negatively regulates miR-542-3p and miR-542-3p negatively regulate ILK in CRC, which was testified in our research.

A meta-analysis had evaluated the impact of *CASC9* on the prognosis and clinicopathological features of patients with cancer, found that a higher *CASC9* signified a severe invasion, and high *CASC9* meant lower overall survival rate [21]. In nasopharyngeal cancer, *CASC9* promotes cell proliferation by hypoxia-inducible factor-1α. In esophageal squamous cell carcinoma, *CASC9* downregulates programmed cell death 4 protein. *CASC9* contributes to cellular proliferation by upregulating the glucose transporter 1 gene in laryngeal carcinoma. *CASC9* also regulates cell cycle progression in the G1 phase via cyclin D1 in lung adenocarcinoma. In glioma cancer cells, *CASC9* magnifies its oncogenic potential via exerting the signal transducer and activator of transcription 3 transcription factor by miR-519d. *CASC9* positively regulates checkpoint kinase1 by miR-195/147 cluster in breast cancer cells. *CASC9*/miR-758-3p/*LIN7A* pathway was identified in ovarian cancer progression. *CASC9* expresses highly by miR-125b/ neuregulin-1 in hemangioma, colon, lung, and gastric cancers, etc. [21].

In CRC, *CASC9* is significantly upregulated, and higher *CASC9* is associated with tumor progression and poor outcomes by polyadenylation specificity factor subunit 3 and *TGFβ2* mRNA [4, 18]. Silencing *CASC9* leads the reduced CRC cell to proliferate and migrate through AKT/mTOR/EMT signaling [5]. Knock-downing *CASC9* inhibits proliferation via miR-576-5p/AKT3 and promotes apoptosis [4].

In our study, we also confirmed that *CASC9* expresses unusually highly in clinical specimens and CRC cells. Down-regulating *CASC9* could inhibit CRC cells from proliferating and invading. LncRNA *CASC9* negatively regulates miR-542-3p in a targeted manner, and up-regulating miR-542-3p inhibits CRC cells from proliferating and invading. However, ILK is a driver mutation in CRC. Thus, the effect of *CASC9* plus miR-542-3p is the same as ILK in CRC.

MiR-542-3p inhibits invasion and migration of esophageal cancer by repressing *OTUB1* [22], suppresses the proliferation of osteosarcoma via targeting Smad2 [23], inhibits progression of oral squamous cell carcinoma by impeding ILK/TGF-β1/Smad2/3 [24].

In CRC, miR-542-3p can impede tumor promotion via *TUG1* [25], and inhibit the proliferation, migration, and invasion through *OTUB1* [8], cortactin [19], or PI3K/AKT signaling [20].

There are some limitations in this study. First, we verified the inhibition of downregulating lncRNA *CASC9* and upregulating miR-542-3p only at the gene level, neglecting the protein level, otherwise, the results will be more convincing. Second, treating silencing lncRNA *CASC9* as a method for curing CRC has not yet been seen in a clinical study. It can be known from basic experiments that overexpression of ILK on down-regulated lncRNA *CASC9* or up-regulated miR-542-3p is expected to become therapy in the future. Finally, we don't know whether there are other mechanisms to impact this pathway, which still need to be studied in the future.

## Conclusions

In our study, we demonstrated that lncRNA *CASC9* expresses highly in CRC, and downregulating it could inhibit cellular proliferation and invasion. Once upregulated, miR-542-3p played an inhibitory role. LncRNA *CASC9* did negatively regulate miR-542-3p in a targeted manner, and ILK was the target gene of miR-542-3p. The overexpression of ILK could significantly restore the proliferation inhibited by downregulated lncRNA *CASC9* or upregulated miR-542-3p. Thus, we predict that lncRNA *CASC9*/miR-542-3p/ILK may become a feasible mechanism for curing CRC in the future.

## Supporting information

**S1 Raw images.**
(PDF)

**S1 File. Original data.**
(PDF)

**S2 File. Images of correlation between CASC9 and ILK from databases.**
(PDF)

## Author Contributions

**Conceptualization:** Heping Zhang, Zongyang Yu.

**Data curation:** Heping Zhang, Jingfang Wang, Taoyuan Yu, Zongyang Yu.

**Formal analysis:** Heping Zhang, Jingfang Wang, Taoyuan Yu, Jun Lu, Zongyang Yu.

**Funding acquisition:** Heping Zhang, Jingmin Wang.

**Investigation:** Heping Zhang, Jingfang Wang, Taoyuan Yu, Zongyang Yu.

**Methodology:** Heping Zhang, Jingfang Wang, Taoyuan Yu, Jingmin Wang, Jun Lu, Zongyang Yu.

**Project administration:** Heping Zhang, Jingfang Wang, Jingmin Wang, Jun Lu, Zongyang Yu.

**Resources:** Heping Zhang, Jingfang Wang, Jingmin Wang.

**Software:** Jingfang Wang, Taoyuan Yu.

**Supervision:** Heping Zhang, Jingmin Wang.

**Validation:** Heping Zhang, Jingmin Wang, Zongyang Yu.

**Visualization:** Heping Zhang, Zongyang Yu.

**Writing – original draft:** Heping Zhang, Jingfang Wang.

**Writing – review & editing:** Heping Zhang, Jingfang Wang, Zongyang Yu.

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
