## [Decision Letter · Decision Letter 0]

22 Oct 2021

PONE-D-21-27936Silencing LncRNA CASC9 Inhibits Proliferation and Invasion of Colorectal Cancer Cells by MiR-542-3p/ILKPLOS ONE

Dear Dr. Zhang,

Thank you for submitting your manuscript to PLOS ONE. After careful consideration, we feel that it has merit but does not fully meet PLOS ONE’s publication criteria as it currently stands. Therefore, we invite you to submit a revised version of the manuscript that addresses the points raised during the review process.Please revise the manuscript in response to all concerns of Reviewer 1; The manuscript is poorly written.   Please have the whole manuscript edited well;Please analyze the correlation of expression levels of LncCASC9, miR-542 and/or ILK in human tumor tissues using data base;Please analyze expression levels of LncCASC9, miR-542 and ILK in mouse tumor tissues;miR-542 has many targets as ILK.  Please identify another target and show with ILK.Please submit your revised manuscript by Dec 06 2021 11:59PM. If you will need more time than this to complete your revisions, please reply to this message or contact the journal office at plosone@plos.org. Please include the following items when submitting your revised manuscript:A rebuttal letter that responds to each point raised by the academic editor and reviewer(s). You should upload this letter as a separate file labeled 'Response to Reviewers'.A marked-up copy of your manuscript that highlights changes made to the original version. You should upload this as a separate file labeled 'Revised Manuscript with Track Changes'.An unmarked version of your revised paper without tracked changes. You should upload this as a separate file labeled 'Manuscript'.

We look forward to receiving your revised manuscript.

Kind regards,

Bing-Hua Jiang, Ph.D.

Academic Editor

PLOS ONE

Journal Requirements:

2. Please provide additional details regarding participant consent. In the Methods section, please ensure that you have specified (1) whether consent was informed and (2) what type you obtained (for instance, written or verbal). If your study included minors, state whether you obtained consent from parents or guardians. If the need for consent was waived by the ethics committee, please include this information.

Reviewers' comments:

Reviewer's Responses to Questions

**Comments to the Author**

1. Is the manuscript technically sound, and do the data support the conclusions?

Reviewer #1: Yes

2. Has the statistical analysis been performed appropriately and rigorously? 

Reviewer #1: Yes

3. Have the authors made all data underlying the findings in their manuscript fully available?

Reviewer #1: Yes

4. Is the manuscript presented in an intelligible fashion and written in standard English?

Reviewer #1: Yes

5. Review Comments to the Author

Reviewer #1: The authors showed that silencing long noncoding RNA cancer susceptibility candidate 9 with miR-542-3p/integrin-linked kinase could inhibit proliferation and invasion of colorectal cancer cells. The experiments were well conducted, applying different methods to support the effect of silencing lncRNA CASC9 in colorectal cancer.

Suggestions:

1. Please check the English proficiency of the manuscript. There are some grammatical mistakes. For example, “We firstly detected the expression of lncRNA CASC9, then evaluated the effect of lncRNA CASC9 on CRC, and how miR-542-3p/ILK to affect lncRNA CASC9.” in the study design would be better changed to “and explored how miR-542-3p/ILK affects lncRNA CASC9.”

2. There should be a space between the number and the unit. For example, it should be “8 μg/ml”, “0.5 μg/ml”, “1 h” and “30 μg” but not “8μg/ml”, “0.5μg/ml”, “1h” and “30μg”.

3. It is recommended that the method and the test are stated in full in the subtitle, e.g. "EdU assay for cell proliferation".

4. The reference should be provided for “IHC assay was performed as described previously using supersensitive horseradish peroxidase immunohistochemistry kit (Sangon Biotech, Shanghai, China) per manufacturer’s instructions……” in “Immunohistochemistry”.

5. For the p value, please unify the case of “p” in the statistical analysis and result.

6. For Fig.4b and Fig.6b, “Dual luciferase report assay revealed that compared to 3 control subsets, the activity in Mut significantly reduced” did not match with the data shown in the figures. Please check them.

7. There should be a scale bar to show the size of the cells for EdU and transwell assay in figures.

6. PLOS authors have the option to publish the peer review history of their article (what does this mean?). If published, this will include your full peer review and any attached files.

Reviewer #1: No

---

## [Author Response · Author response to Decision Letter 0]

3 Dec 2021

We greatly appreciate this opportunity to resubmit a revised version of our manu-script, and hope that our attention to these very constructive comments, are found to be responsive and received favorably by you and your expert reviewers.

---

## [Editor Report · Decision Letter 1]

10 Mar 2022

Silencing LncRNA CASC9 Inhibits Proliferation and Invasion of Colorectal Cancer Cells by MiR-542-3p/ILK

PONE-D-21-27936R1

Dear Dr. Zhang,

We’re pleased to inform you that your manuscript has been judged scientifically suitable for publication and will be formally accepted for publication once it meets all outstanding technical requirements.

Kind regards,

Bing-Hua Jiang, Ph.D.

Academic Editor

PLOS ONE
---

## [Editor Report · Acceptance letter]

5 Apr 2022

PONE-D-21-27936R1 

Silencing LncRNA CASC9 Inhibits Proliferation and Invasion of Colorectal Cancer Cells by MiR-542-3p/ILK 

Dear Dr. Zhang:

I'm pleased to inform you that your manuscript has been deemed suitable for publication in PLOS ONE. Congratulations! Your manuscript is now with our production department. 

Kind regards, 

on behalf of

Dr. Bing-Hua Jiang 

Academic Editor

PLOS ONE